# Incorporation of Cement Bypass Dust in Hydraulic Road Binder

**DOI:** 10.3390/ma14010041

**Published:** 2020-12-24

**Authors:** Nadezda Stevulova, Julius Strigac, Jozef Junak, Eva Terpakova, Marian Holub

**Affiliations:** 1Faculty of Civil Engineering, Institute of Environmental Engineering, Technical University of Kosice, 042 00 Kosice, Slovakia; jozef.junak@tuke.sk (J.J.); eva.terpakova@tuke.sk (E.T.); marian.holub@tuke.sk (M.H.); 2Povazska Cement Factory, 018 63 Ladce, Slovakia; strigac.j@pcla.sk

**Keywords:** hydraulic road binder, cement, limestone, granulated blast furnace slag, bypass dust

## Abstract

This article describes utilization of a cement kiln bypass dust utilization as an added component in a hydraulic road binder. Three experimental binder mixes (BM1–BM3) with variation in the composition of the main constituents (cement clinker, ground limestone and ground granulated blast furnace slag) and constant content of bypass dust (10%) were prepared under laboratory conditions. The properties of binder constituents, fresh experimental binder mixes and hardened specimens were tested according to STN EN 13282-2 for a normal hardening hydraulic road binder. The physical and chemical properties of all binder mixes (fineness: +90 µm ≤ 15 wt.%; SO_3_ content: <4 wt.%) met the standard requirements. The bypass dust addition led to an increase in the water content for standard consistency of cement mixes (w/c = 0.23) and to a shortening of the initial setting time for two experimental blended cement pastes (BM1 and BM3) compared with the value required by the standard. Only BM2 with the lowest SO_3_ content (0.363 wt.%) and the highest percentage of granulated blast furnace slag (9.5 wt.%) and alkalis (Na_2_O and K_2_O content of 5.9 wt.%) in the binder mix met the standard value for the initial setting time (≥150 min). The results of compressive strength testing of experimental specimens after 56 days of hardening (59.2–63.9 MPa) indicate higher values than the upper limit of the standard requirement for the N4 class (≥32.5; ≤52.5 MPa).

## 1. Introduction

The sustainable construction industry focuses on the production of new environmentally friendly solutions connected with the replacement of conventional materials. Cement production is an intensive consumer of power, natural nonrenewable raw materials, and fossil fuels in a high-temperature process. The contribution of the cement manufacturing sector to the total global anthropogenic CO_2_ emissions is approximately 8% [1,2]. These environmental impacts are associated with CO_2_ emissions from clinker production (especially the thermal decomposition of calcium carbonate) and combustion of fossil fuels. CO_2_ is a greenhouse gas that contributes to global warming and related climate changes. Therefore, cement production urgently needs to apply the principle of sustainability and to use supplementary cementitious materials and substitutes for conventional fossil fuels when heating cement kilns through the use of alternative fuels with adequate calorific values. Recycling of appropriate products, such as waste and/or by-products, has become an attractive alternative to their disposal in accordance with the waste management hierarchy, as well as with European Union (EU) policies and Agenda 21 targets related to sustainable development. Benefits related to the usage of alternative fuels are associated with a reduction in environmental and economic burdens. Different types of gaseous, liqud, and solid wastes (hazardous and non-hazardous) are used as alternative fuels in cement kilns, such as waste gases from refinery, landfill, and pyrolysis, waste oils and fats, wax suspensions, chemical waste, asphalt slurry, industrial sludge, municipal sewage sludge, agricultural waste, rubber and tires, plastics, and textiles [3]. At present, attention is being paid to the replacement of fossil fuels with carbon neutral materials (agricultural biomass, municipal solid waste, or meat and bone from animal-based meals) [4]. The potential of these alternative fuels is related to their suitability for the combustion process and ensuring conditions of high-energy efficiency [5]. This strategy of the cement industry based on the partial substitution of traditional fossil fuels with alternative fuels for cement clinker production is of high importance and an attractive alternative to nonrenewable fossil fuels [4]. However, the use of alternative fuels must abide by the rules related to a reduction in CO_2_ emissions during the burning process in a cement kiln [6]. An increasing proportion of alternative fuels in the combustion process led to a need to implement bypass technology in production, thereby ensuring the removal of exhaust gases from the cement rotary kiln. Bypasses with different configurations are so far the only possible method of reducing the amount of chlorides in the cement kiln atmosphere [7,8]. Large quantities of particulate material present in the flue gases are removed from the kiln by air pollution control devices. Cement kiln dust (CKD), also known as bypass dust (BPD), is a finely dispersed particulate material, composed of oxidized, anhydrous, micron-sized particles accumulated in the dust collectors such as cyclones, bag houses, or electrostatic precipitators during the high-temperature Portland cement clinker process. This by-product containing fine particulates of unburned and partially burned raw materials is rich in alkalis, chlorides, and sulfates and may also contain heavy metals [9]. Alkaline chlorides are introduced to the cement kiln through the fuel. Thus, the chemical composition of BPD is determined by the clinker burning technology, the type of alternative fuel burned, the raw materials, and the exhaust gas removal system [10,11]. According to the Environmental Protection Agency (EPA) [12], BPD is considered as a potentially hazardous waste due to its caustic and irritating nature. Due to its high alkali content exceeding the cement clinker standard tolerances, the BPD by-product is not possible to return to the feedstock of clinker manufacturing. The amount of BPD generated per ton of clinker produced is equal to approximately 15–20% (by mass) [13,14,15]. The total volume of BPD generated in the Slovak Cement Factories is about 6% of the total annual production of clinker, which currently ends up in an on-site landfill for hazardous waste. High amounts of BPD per year lead to the high cost of collection, transportation, and landfilling due to a lack of adequate methods for its reuse and recycling. The existence of this waste and its growing production pose a serious environmental problem. Therefore, it is mandatory to minimize this by-product of cement production, or to handle it in a sustainable manner.

Literature sources indicate that BPD can be reused in a number of different ways for various purposes. As reported in many papers, the most common applications of BPD are in soil consolidation and stabilization [10,16]. BPD addition can lead to an increase in the shear strength of a soil and/or control the shrink-swell properties of a soil, thus improving the load-bearing capacity of a sub-grade to support pavements and foundations [17]. BPD combined with fly ash was successfully used for stabilizing road beds and, especially, clay soils [18]. Among modern engineering applications of BPD, its significant application potential and widespread utilization is in the construction sector. Mostly, BPD is applied as a partial replacement of cement in the production of mortar/concrete [10,19,20,21,22], cement bricks [23], concrete paving blocks [24,25], asphalt pavement/concrete [26], and sand sidewalks [9]. The presence of clinker phases and free lime in BPD may be responsible for its binding properties. Many papers reported the effect of used BPD on the different properties of mortar, and concrete [10,21,22,27]. There were a fewer older works devoted to the characterization of BPD and its influence on the properties, the results of which were summarized in [28]. Wide variation in the chemical composition of CKDs limits their potential application as a sustainable binder component in concrete. However, BPD application in suitable amounts is not adversely affecting their properties. As presented in [19], concretes with lower percentages of BPD (about 5%) reach compressive strength, flexural strength, toughness, and freezing and thawing resistance comparable to the traditional concrete.

The presence of free CaO and significant amounts of soluble sodium and potassium compounds (chlorides and sulfates) in BPD causes its high alkalinity [29]. Thanks to BPD’s alkaline nature, it can be used as an alkali activator of supplementary cementitious materials [30,31,32]. In recent years, many scientific advances in the preparation technology and the insights into the performances of new binders with a geopolymer matrix developed by blending BPDs with the fly ash and/or granulated blast furnace steel slag (GGBS) have performed [15,33,34]. As shown in [35], BPD addition enhances the geopolymerization reaction. The alkaline-modified alternative constituents improve the fresh mixture properties of the cement mortar/concrete and contribute to a reduction in the environmental impact. Successful utilization of BPD for fly ash and GGBS activation in ternary/quaternary cement system, leading to its higher strength, depends on the dissolution rate of additives, the alkalinity of the reacting system, and the existence of the optimum free lime and sulfate content [32,36]. In accordance with paper [37], the alkali activated cements are characterized by more durable geopolymer matrix and lower carbon footprint than ordinary Portland cement. Heikal et al. [31] reported on more durable binder with optimized mix composition of GGBS, BPD, and/or micro-silica, where 20% BPD amount was used.

The incorporation of BPD in cement-blended material represents the most advanced strategy for an environmentally friendly waste management solution for BPD. Taking this aspect of BPD into account, alternative cement binder materials are becoming an important research topic.

Limited studies were aimed at evaluating the performance of cement mortars incorporating BPD and supplementary cementitious materials for infrastructure applications are currently available. Evaluation of the physico-mechanical and durability-related properties of concrete paving blocks containing BPD as a partial replacement or as an addition to cement indicated that up to 40–60% BPD could be used for producing environmentally friendly paving blocks for traffic applications [24]. As presented in [38], cement binders containing BPD can significantly improve the properties of the sub-grade.

BPD can be a constituent of cement binder with a high proportion of mineral components. A representative example is the hydraulic road binder. European standard requirements for a normal hardening hydraulic road binder refer to the physical and chemical properties of binder mixture, as well as the mechanical properties after 56 days of hardening. To the author’s knowledge, there are few reports on the use of BPD in quaternary cement mixes but with a different composition for this purpose. Therefore, this study highlights its significance and originality.

Scientific hypothesis was supported by positive initial results that BPD containing a combination of the activators (CaO, and different forms of alkalis) can potentially result in a new binding system with favorable mechanical properties when used in conjunction with GGBS.

The main objective of the research was a comprehensive solution for the use of BPD in preparing experimental mixes for an eco-friendly normal hardening hydraulic road binder, with the key properties of fresh cement pastes and hardened specimens tested according to the standard in [39]. Another partial objective is to contribute to the available knowledge about adoption of alternative hydraulic road binder material that can significantly improve its performance with BPD addition for geopolymer formation in the binder system, and environmental protection. Therefore, the novelty of our research is in utilizing and evaluating the properties of a hydraulic road binder containing cement clinker and limestone as the major binder constituents with a portion of supplementary cement material (GGBS) and in combination with the additional substance of BPD. The scientific problem lies in understanding the influence of the physico-chemical characteristics of BPD on its performance as an effective binder component, and on the properties of fresh mixes and hardened mortar specimens. The outcome of this research is expected to provide critical information on the development of mortar properties in accordance with the standard requirements.

The choice of this additional binder constituent was motivated by the improvement in properties of the binder mix (mainly workability) in accordance with the standard in [39]. Three experimental mixes with variations in the main constituents (cement clinker and limestone) and GGBS were prepared. The BPD amount in the experimental mixtures was constant at 10% of the total weight in accordance with the standard recommended content of an additional binder constituent in normal hardening road binder. An additional aim of this research was to assess the critical properties, including the fineness, chemical and phase composition of dry binder mixes, the behavior of fresh mixes (initial setting time, consistency), and compressive strength of hardened bodies, and to interpret the obtained results. To achieve these aims, all standard requirements with respect to the properties of a normal hardening road binder were tested. The experimental investigations were organized in two phases. In the first, the properties of BPD and experimental binder mixtures were assessed in terms of their compliance with standard requirements for their mix composition. In the second, the resulting properties of fresh pastes and specimens hardened under normal conditions were evaluated.

## 2. Materials and Methods

### 2.1. Input Materials of Hydraulic Binder Mix

The source materials for the main constituents of the hydraulic road binder were as follows: Portland clinker, mineral additives in the form of ground limestone, and ground granulated blast furnace slag (GGBS). Cement kiln bypass dust (BPD) collected right at the cold end of the kiln in the cement plant and at the same time as the clinker was used as an additional constituent in the experimental binder mixtures. The chemical composition of the hydraulic binder constituents determined by X-ray fluorescence spectroscopy (XRF; SPECTRO iQ II XRF spectrometer, Ametek, Unterschleissheim, Germany) is given in Table 1. Na_2_O content was determined using flame photometry (Flame Photometer PFP7, JENWAY, Staffordshire, UK).

X-ray diffraction analysis was used for determination of the mineralogical composition of binder components (BRUKER AXS D8 Advance Diffractometer, Billerica, MA, USA). The main mineral phases identified in the binder constituents are listed in Table 2.

The particle size analysis of all hydraulic road constituents was performed separately using a laser granulometer (Mastersizer 2000; Malvern Instruments Ltd., Malvern, UK), because the standard sieving as one of the oldest particle sizing methods is widely used only for relatively large particles. The results of particle size analysis are shown in Table 3. The values of D10, D50, D90, specific surface area (S) and surface weighted mean (SM) describing the integral characteristics of the particle size distribution are summarized in Table 4.

The D values refer to the distribution of constituents on the basis of their diameter (e.g., D50 denotes that half of the constituents are above and half are below this diameter).

### 2.2. Hydraulic Binder Mix Composition

According to the standard in [39], the content of the main binder constituents should be higher than 10% by weight, with the additional component not exceeding 10% of the total weight of the mixture in a normal hardening hydraulic road binder. Formulations for experimental mixtures were designed according to the recommended range of constituents for a normal hardening hydraulic binder according to the DoroCem product data sheet [40] with a constant content of BPD (Table 5).

The contents of individual constituents in the experimental binder mixtures BM1–BM3 are given in Table 6. BM1 features an average content of constituents (clinker, limestone, and GGBS) in the recommended range, whereas BM2 and BM3 feature the maximum and minimum levels of the recommended range, respectively.

### 2.3. Preparation of Blended Cement Pastes

The introduction of BPD into the cement mixture necessitated an increase in the water required to obtain a cement paste of standard consistency. Due to the dry mixture and poor process ability of cement mixtures with a water coefficient (w/c) = 0.5, we had to optimize this ratio to a value of 0.23. The reference mix and experimental binder pastes were prepared according to the standard in [41] by mixing of the dry binder components with deionized water in a standard E093 mixer (MATEST, Treviolo, Italy) at low speed for 120 s. The mix was then stopped for 90 s to remove any paste adhering to the wall and bottom of the bowl, as well as to stirrer, using a plastic scraper. Mixing then continued at a higher speed for 120 s.

A steel mold for three bodies with dimensions of 40 × 40 × 160 mm each was filled with fresh mix, and a laboratory vibrating table (MATEST, Treviolo, Italy) equipped with a motor generating 3000 vibrations per minute was used for compaction of the cement mortar specimens. Vibration continued until no air bubbles were present on the mortar surface and the body surface was relatively smooth with a glossy appearance. The vibrated fresh paste mixtures were stored in a humid environment (covered with polyethylene terephthalate (PET) foil) and, after 48 h, the bodies were demolded and placed in an aqueous medium. Deionized water (from the laboratory of Faculty of Civil Engineering of TU in Kosice) was used for the processing of mixes according to [42]. Mixing and hardening processes were carried out in laboratory conditions (temperature of 23 °C; relative humidity of 55–65%).

### 2.4. Testing Procedures

#### 2.4.1. Fresh Mix Properties

The setting times and the consistency of cement paste are key factors affecting the technological processes of mortars and concrete production. The initial setting time point was determined using the probe penetration method. The initial setting time for all fresh experimental cement pastes was determined as the time elapsed from zero to the moment when the distance between the cylindrical steel needle and the pad reached 6 ± 3 mm according to the standard in [41] using a Vicat apparatus (ELE International, Sheffield, UK).

The consistency of experimental fresh binder mixes was determined using a flow table test in accordance with [43] for freshly mixed mortars including minerals binders, whereby the mean diameter of a test sample placed on a flow table was measured before being impacted vertically after release of a standard slump cone.

#### 2.4.2. Properties of Hardened Specimens

The bulk density of the test specimens was evaluated according to [44]. The compressive strength of experimental cement specimens partially substituted with alternative binder constituents was determined according to [45] after 28, 56 and 90 days of hardening (ADR ELE 2000, International Ltd., UK). The resulting parameter values were the average of six measurements. The standard requirements for mechanical properties are provided in Table 7 as characteristic values of a normal hardening hydraulic road binder.

## 3. Results and Discussion

### 3.1. Properties of Dry Experimental Hydraulic Road Binder Mixes

#### 3.1.1. Fineness

The fineness of all hydraulic road constituents was determined separately due to the fact that we did not have a suitable homogenizer for preparing a dry mixture. As can be seen from Table 3 and Table 4, the finest component was BPD, with particles no larger than 70 μm and a D50 value of 4.13 μm. In contrast, the particle size distributions of ground limestone, clinker, and GGBS featured larger D10, D50 and D90 values. The specific surface area calculated from particle size analysis of the individual binder constituents ranged from 1.10 to 2.88 m^2^·g^−1^, which correlated with the surface weighted mean diameter values in the range of 2.16–5.47 μm. The surface characteristics of cement binder components related to their fineness, representing the surface available for hydration, were of following order: BPD > limestone > clinker > GGBS.

The proportion of particles larger than 90 μm in the binder constituents, as shown as shown in Table 8, met the standard requirement for the fineness of a normal hardening hydraulic road binder (≤15%).

#### 3.1.2. Chemical and Phase Composition

As specified in [39], the standard requirements for the chemical properties of a normal hardening hydraulic road binder are a function of the sulfate content, expressed as SO_3_ percentage by weight, which should not exceed 4%. The data in Table 1 confirm that the sulfate content in BPD was 11.13 wt.%, while that in GGBS was 4.03 wt.%. However, the calculated SO_3_ contents for the experimental hydraulic road binder mixes were lower than 4 wt.% (Table 9).

Table 10 lists the CaO and SiO_2_ contents and CaO/SiO_2_ ratio in the experimental binder mixtures. A CaO/SiO_2_ ratio of approximately 3 ensures the formation of calcium silicate hydrates (C-S-H). The Al_2_O_3_ content in the experimental cement mixtures ranged from 3.09 to 4.17 wt.%, thereby facilitating cement hydration to increase the strength of specimens (Table 11). The content of soluble alkalis, expressed as the sum of Na_2_O and K_2_O in the experimental mixtures (about 5.79–5.81 wt.%), is required for the geopolymerization reaction. Lastly, free CaO or its hydrated form present in the clinker and BPD is able to participate in the pozzolana reaction with SiO_2_ and Al_2_O_3_. The contents of the individual oxides did not differ greatly in the experimental mixes.

A study of the mineralogical composition of the individual binder road constituents confirmed the presence of phases commonly occurring in these materials (Table 2). The C_3_S phase was a major component in cement clinker (70.3 wt.%), whereas the C_2_S and C_4_AF contents ranged from 7.8 to 9.3 wt.%. The percentage of the orthorhombic phase of C_3_A was 6.4 wt.%, while the cubic phase represented a lower content (2.7 wt.%). The content of portlandite was about 1%, with remaining phases having an even lower content.

The major phase in limestone was calcite, while minor crystalline phases were represented by dolomite and quartz.

GGBS as a ground by-product of iron and steel from a blast furnace was represented by a Ca–Mg–Al silicate glassy phase. CaO in GGBS tends to form disordered calcium/magnesium aluminosilicate, which is largely responsible for its reactivity during alkali activation [46]. The high pH of the alkaline activator promotes dissolution of GGBS, which drives the chemical reaction and strength development [47]. As is known [48], Al_2_O_3_ and MgO also play a role in the formation of alkali-activated GGBS paste.

### 3.2. Properties of Fresh Cement Mixes

#### 3.2.1. Initial Setting Time

Due to BPD fineness, its amount of 10 wt.% in the experimental binder mix caused an increase in water demand compared to traditional cement paste, in accordance with [49]. An average value of the water requirement for all mixes was 23 wt.%. The initial setting times for the experimental fresh binder mixes determined as a function of the depth of needle penetration are given in Table 12. For the reference cement clinker paste, an initial setting time of 120 min was found, related to the cement class 32.5 specified in [50]. It was observed that the composition of experimental blended cement pastes BM1 and BM3 resulted in a shortening of their initial setting times compared to the standard [39] for a normal hardening hydraulic road binder (≥150 min). These results are consistent with findings in works [19,51,52], where 10% BPD replacement of cement led to a decrease in the initial setting time. The lowest initial setting time value was identified for experimental blended cement paste BM3 (120 min), followed by experimental binder mixture BM1 (130 min). The shortening of the initial setting time can be caused by the content of alkalis that activates the hydration of clinker in binder mixes [53]. The decrease in the initial setting time with the higher water requirement for normal consistency in binder mixes indicates hydraulic properties of BPD.

The initial setting time of BM2 (150 min) confirmed that only this binder mix met the standard value, likely related to the lowest SO_3_ content (0.363 wt.%). Furthermore, the differences in the initial setting time could have been caused by the fineness of binder constituents in accordance with their inverse relationship [54].

#### 3.2.2. Consistency

As shown in Table 13, all experimental cement blended pastes, including the reference paste, reached mean spill diameter values in the range 146–164.5 mm, classifying them as plastic mortars (140–200 mm).

Experimental pastes BM1 and BM2 showed a smaller spill diameter compared to the reference mix, whereas BM3 was more plastic with a higher spill diameter. BM1 was closest to the reference spill diameter value; however, the difference in values between BM1 and BM2 was only 2 mm. The behavior of the binder mixes during the spill test is illustrated in Figure 1. In accordance with [55], the water to binder ratio, rate of hydration reactions and fineness of the binder materials mainly determines the consistency. It should be emphasized that the standard for a normal hardening hydraulic road binder [39] does not require a determination of consistency.

### 3.3. Properties of Hardened Specimens

#### 3.3.1. Bulk Density

The bulk density of the specimens after 28, 56, and 90 days of hardening is presented in Table 14. The bulk density of the reference bodies was lower than that of the mortars based on experimental hydraulic road binder mixes (BM1–BM3) depending on the hardening time. The results in Table 14 indicate that experimental binder mixes exhibited bulk density values in the range of 2100–2250 kg·m^−3^, consistent with the ordinary mortars (1600–2300 kg·m^−3^) [44].

#### 3.3.2. Compressive Strength

The compressive strength after 28, 56 and 90 days of hardening for experimental binder mixes prepared at a constant water-to-binder ratio of 0.23 with varied the binder constituents and constant addition of BPD is presented in Figure 2. The variation in the measured compressive strength values of each hardened mortar sample was about ±10%. The lowest variance of the strength parameter (±5%) was recorded for the BM1 sample with the highest values of compressive strength after hardening compared to other mortar samples.

A comparison of the values (shown in Figure 2) demonstrates that the compressive strength increased with hardening time. All experimental specimens after 56 days of hardening reached higher strength values (59.2–63.9 MPa) than the reference sample (51.90 MPa). The BM3 sample achieved the highest value of compressive strength, where the main binder constituents in the mixture (clinker, limestone, and GGBS) were at the minimum level of the recommended range for the DoroCem product. High compressive strength values (60–69 MPa) of quaternary mixes consisting of 30 wt.% Portland cement, 45–50 wt.% fly ash, 5–10 wt.% BPD, and 15–18.5 wt.% GGBS after 28 days of hardening were achieved [56].

The relative compressive strength calculated as the ratio of the compressive strength value of experimental specimens BM1–BM3 and the reference sample (Table 15), showed an increasing trend of 14–23% 56 days.

The compressive strength after 56 days of hardening is a key factor in meeting standard requirements for the mechanical properties of a normal hardening hydraulic binder. As specified in [39], for class N 4 (Table 2), this value should be in the range of 32.5 to 52.5 MPa. The measured values of compressive strength for experimental specimens after 56 days of hardening (59.2 to 63.9 MPa) were 13–21.6% higher than the upper limit of the range for class N4. The BM3 sample with the highest CaO/SiO_2_ ratio (3.11 wt.%) achieved the highest increase in compressive strength (21.6%) compared with the reference specimen.

The high strength found for the experimental binder specimens can be attributed to the hydration of the clinker as well as the geopolymeric compounds formed in the binder mix consisting of limestone, GGBS, and BPD. As is known, hydration involves a complex group of reactions causing changes in the chemical and physico-mechanical properties of the system, particularly in the setting and hardening of the binder mix. This process is sensitive to many factors of a physical and chemical nature. One of the critical factors in attaining the final strength of a binder in allowable time is its particle size distribution. A smaller particle size results in a greater surface area-to-volume ratio, with more area available for water–cement and binder particle interactions per unit volume. Therefore, finer binder materials react faster with water molecules, resulting in a higher rate of strength development.

On the other hand, the chemical composition of the binder also plays an important role in the formation of various hydrated phases. The clinker composition is of special relevance, whereby the presence of free CaO in the binder mix is very important for the formation of the main hydration products of C-S-H and C-S-A-H. The strength of specimens was significantly affected by the relatively high content of free CaO in BPD capable of rapid hydration to Ca(OH)_2_ and a subsequent pozzolanic reaction with the active components of SiO_2_ and Al_2_O_3_ present in GGBS. Sodium and potassium oxides present in the binder mix are important for the formation of silicon-based geopolymers (Men[-(SiO_2_)_z_-AlO_2_]_n_.wH_2_O; Me = Na^+^ or K^+^) as well as for the creation of n1Na_2_O·n2CaO·n3SiO_2_·n4H_2_O [57].

Calcium monosulfoaluminate hydrate (AFm phase) and ettringite (AFt phase) are commonly present in cements. The crystalline structure of AFm phases allows the incorporation of one monovalent anion or half of a divalent anion in the interlayers. In contrast with AFt phases, AFm phases are characterized by a better ability to bind chloride ions. It is known that the chloride- binding capacity in cement paste can also contribute to the formation of specific hydrated phase such as Friedel’s salt (3CaO·Al_2_O_3_·CaCl_2_.10H_2_O) [58]. The presence of sulfate ions in the binder mix leads to the formation of Kuzel’s salt in which chloride ions are partially substituted by sulfate ions (3CaO·Al_2_O_3_·0.5CaSO_4_·0.5CaCl_2_.11H_2_O) [59], while ferro-aluminosilicate with repeating units of (-Fe-O-Si-O-Al-O-) can also be formed. There is also a possibility to form further other hydration products including calcium aluminate-ferrite hydrate—Ca_2_(Al,Fe)(OH)_6_—and hydrotalcite Mg_6_Al_2_CO_3_(OH)_16_·4H_2_O) [60].

The mechanism of underlying alkali-activation of commercial granulated blast furnace slag is still not fully understood in the existing literature. The composition and mineralogy of raw pozzolans are critical in the formulation of alkali-activated geopolymer materials. The alkali-activation of GGBS products mixed with 3.5–5.5 wt.% sodium hydroxide or water glass yields a low-basic, highly amorphous C-S-H gel product possessing high aluminum content [61]. BPD is an excellent alkaline activator enhancing the dissolution of Si- and Al-containing GGBS [31]. It is assumed that the Si–O-Si, Al–O–Al and Al–O–Si bonds in the aluminosilicate, silicate, and/or aluminate solids are broken first and the dissolution into alumina silicate species and together with alkali cations polymerize and produce the geopolymer network. Consequently, the precipitation of formed hydration products occurs in the form of sodium (or potassium)—aluminosilicate—hydrate similar to natural zeolites (Na-A-S-H or K-A-S-H) in the experimental binder mix [62]. In the final hardening phase of the matrix, excess water exclusion occurs, and a three-dimensional zeolite structure-analogous semi-amorphous phase is formed. In this structure, [SiO_4_]^4-^ and [AlO_4_]^5-^ tetrahedrons are linked by bridging oxygen with the charge imbalance compensated for sodium and potassium cations [63]. These arguments support the above discussion regarding the composition of experimental binder mixes with respect to their mechanical properties.

## 4. Conclusions

According to the evaluation of results obtained from experiments with respect to the standard requirements for a normal hardening hydraulic road binder, the following conclusions can be made:(1)Verifying the designed composition and properties of experimental binder mixes showed the possibility of producing a normal hardening hydraulic road binder using clinker, ground limestone and ground granulated blast furnace slag with an additional component of cement kiln bypass dust not exceeding 10% of the total weight of the mix. The standard requirements for the physical and chemical properties of the experimental binder mixes can be considered fulfilled on the basis of their evaluation, as a function of the fineness, sulfate content, and initial setting time.(1.1)The fineness of the binder mixes met the standard requirement with the proportion of particles larger than 90 μm being ≤15 wt.%.(1.2)The sulfate content expressed as SO_3_ did not exceed the standard limit of 4 wt.% in all experimental binder mixes; in fact, it was 10 times lower.(1.3)The standard requirement for the initial setting time (≥150 min) was only met by the experimental cement blended paste BM2 with the lowest SO_3_ content. The binder mix contained the lowest proportion of clinker and the highest proportion of ground limestone and GGBS among the tested mixes.
(2)The compressive strengths of all experimental specimens reached a higher strength after 56 days of hardening than that specified in the standard STN EN 13282-2 for class N 4 (≥32.5 MPa, ≤52.5 MPa), exceeding the upper limit of the range by 13–21.6%. According to the literature, we hypothesized that the high compressive strength values were due to the resulting structure of geopolymer products.

According to the preliminary results, it can be concluded that the cement kiln bypass dust can be used as an additional component in a normal hardening hydraulic road binder (<10 wt.%). Although the substitution of cement with supplementary cementitious materials (GGBS and BPD) brings both environmental and economic benefits for the cement industry, this research area needs to be expanded to include other important properties to avoid any adverse effects related to the degradation of such a road binder (e.g., volume stability, water absorption, and frost and chemical resistance). Therefore, our further research will focus on a deeper study of the microstructure and durability of this hydraulic binder, thereby contributing to an increase in the complex knowledge of its properties.

## Figures and Tables

**Figure 1 materials-14-00041-f001:**
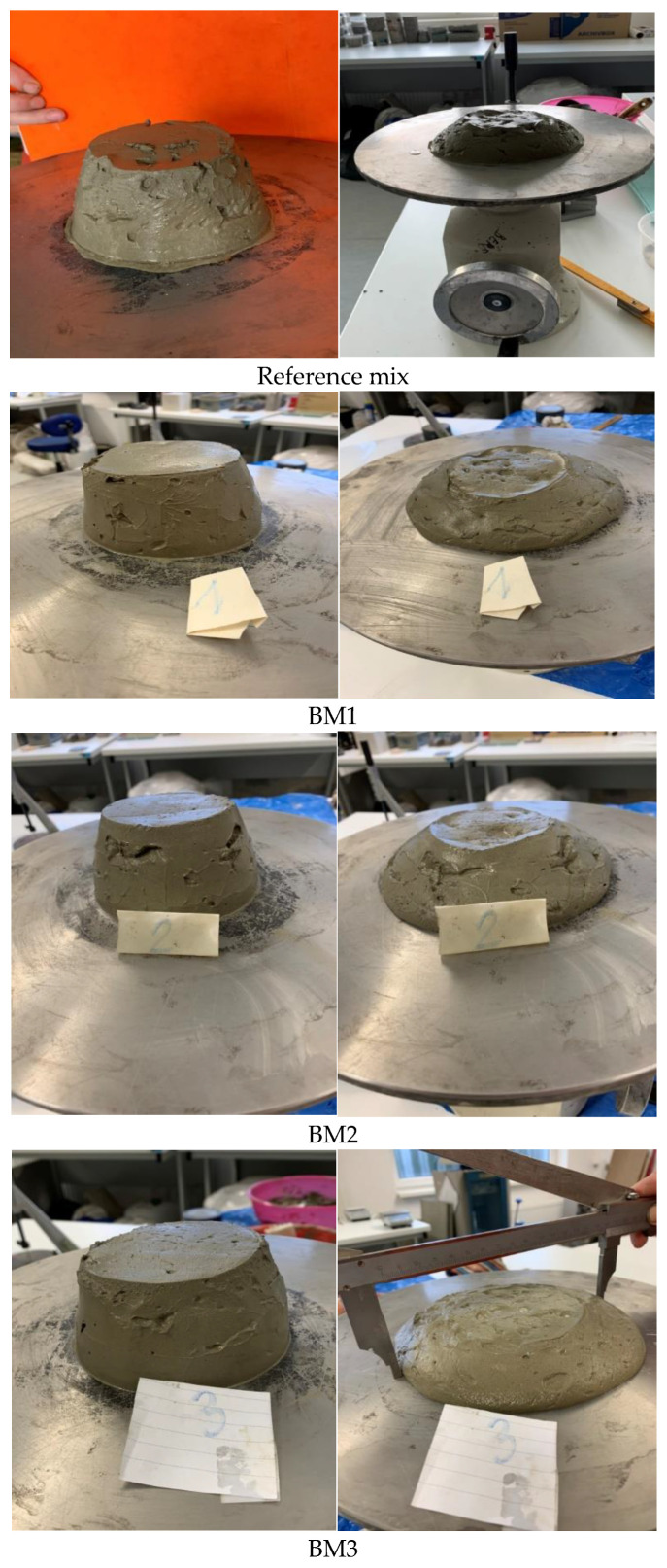
The behavior of the reference cement paste and experimental hydraulic road binder mixes during the spill test.

**Figure 2 materials-14-00041-f002:**
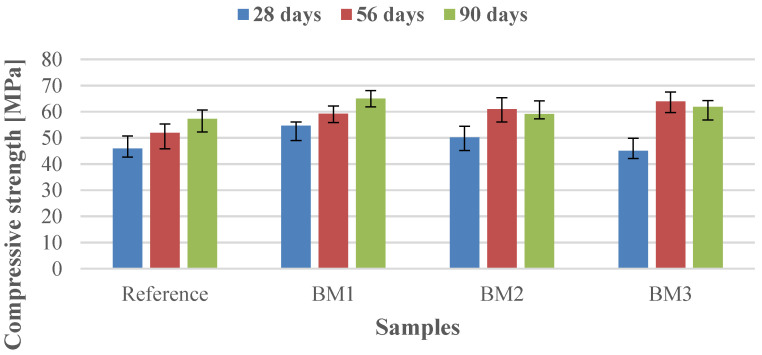
Compressive strength of reference and experimental specimens after 28, 56, and 90 days of hardening.

**Table 1 materials-14-00041-t001:** Chemical composition of hydraulic binder constituents.

Oxides	Clinker	Limestone	GGBS	BPD
		(wt.%)		
Na_2_O	0.24	0.07	0.58	5.44
MgO	1.49	0.57	7.86	0.02
Al_2_O_3_	5.03	1.21	8.11	0.47
SiO_2_	20.67	5.82	42.28	1.92
P_2_O_5_	0.43	0.1	0.03	0.03
SO_3_	0.54	0.05	4.03	11.13
Cl^−^	0.03	0.02	-	33.69
K_2_O	1.05	0.02	0.47	42.83
CaO	65.18	51.95	36.52	7.05
TiO_2_	-	-	0.3	-
MnO	-	0.04	0.78	-
Fe_2_O_3_	2.89	0.51	0.57	0.4
LOI *	0.61	40.01	0.89	0.52

LOI *—loss on ignition.

**Table 2 materials-14-00041-t002:** Mineral phases of binder constituents.

Binder Constituent	Mineral Phases
Clinker	Allite (C_3_S), belite and larnite (C_2_S), C_3_A cubic (pure), C_3_A orthorhombic (Na-doped), colville (C_4_AF), lime (CaO), periclase (MgO), quartz (SiO_2_), arcanite (K_2_SO_4_), portlandite (Ca(OH)_2_)
Limestone	Calcite (CaCO_3_), dolomite CaMg(CO_3_)_2_), quartz (SiO_2_)
GGBS	Melilite (Ca,Na)_2_(Al,Mg,Fe^2+^)(Si,Al)_2_O_7_, merwinite Ca_3_Mg(SiO_4_)_2_
BPD	Sylvite (KCl), arcantite (K_2_SO_4_), halite (NaCl), free lime (CaO), quartz (SiO_2_), anhydrite (CaSO_4_)

**Table 3 materials-14-00041-t003:** Particle size analysis of binder constituents.

Fraction (μm)	Clinker	Limestone	GGBS	BPD
		(wt.%)		
0–1	4.86	16.56	3.42	11.56
1–5	19.72	32.73	14.68	48.52
5–10	16.57	10.12	15.25	22.56
10–20	15.67	10.12	20.97	9.22
20–30	7.63	8.18	14.36	4.19
30–40	9.60	6.46	8.49	2.20
40–70	8.56	7.48	7.58	1.65
70–90	3.76	3.80	4.86	-
+90	3.63	4.55	10.39	-

**Table 4 materials-14-00041-t004:** Values of D10, D50, D90, specific surface area (S), and surface weighted mean (SM) representing integral characteristics of the particle size distribution of binder constituents.

Binder Constituent	D10	D50	D90	S	SM
		(μm)		(m^2^.g^−1^)	(μm)
Clinker	1.98	14.68	119.28	1.37	4.38
Limestone	0.74	5.25	63.29	2.78	2.16
GGBS	2.85	17.37	103.21	1.10	5.47
BPD	0.86	4.13	15.59	2.88	2.08

**Table 5 materials-14-00041-t005:** Recommended composition for a normal hardening hydraulic binder according to the DoroCem product data sheet [40].

Clinker	Limestone	GGBS	BPD
	(wt.%)		
58.5–71.5	16.2–17.8	6.5–9.5	10

**Table 6 materials-14-00041-t006:** Design of recipes with varied constituents in the road binder mixtures.

Recipe	Clinker	Limestone	GGBS	BPD
		(wt.%)		
BM1	65	17	8	10
BM2	63	16.5	9.5	10
BM3	67	17.5	6.5	10
Referential	100	-	-	-

**Table 7 materials-14-00041-t007:** Standard requirements for mechanical properties given as characteristic values.

Compressive Strength	EN 196-1	Requirement			
	Class	N1 *	N2	N3	N4
after 56 Days [MPa]	Value range	2.5–22.5	12.5–32.5	22.5–42.5	32.5–52.5

* A load increase of 400 ± 40 N/s must be used when testing Class 1 test pieces.

**Table 8 materials-14-00041-t008:** Percentage of particles larger than 90 in the individual binder constituents and experimental binder mixes.

Binder Mix	Proportion of Particles +90 μm [wt.%]			
	Clinker	Limestone	GGBS	Binder Mix
BM1	8.86	0.77	0.83	10.46
BM2	8.59	0.8	0.99	10.38
BM3	9.13	0.75	0.68	10.56

**Table 9 materials-14-00041-t009:** Sulfate content in experimental hydraulic road binder mixes.

Binder Mix	SO_3_ [wt.%]
BM1	0.372
BM2	0.363
BM3	0.380

**Table 10 materials-14-00041-t010:** CaO and SiO_2_ content and their ratio in experimental binder mixes.

Binder Mix	Component	Clinker	Limestone	GGBS	BPD	Σ	CaO/SiO_2_
		[wt.%]					
BM1	CaO	42.25	8.83	2.92	0.70	54.70	3.04
	SiO_2_	13.44	0.99	3.38	0.19	18.00	
BM2	CaO	40.95	9.09	3.47	0.70	54.21	2.97
	SiO_2_	13.02	1.02	4.02	0.19	19.25	
BM3	CaO	43.55	8.57	2.37	0.70	55.14	3.11
	SiO_2_	13.85	0.96	2.75	0.19	17.75	

**Table 11 materials-14-00041-t011:** Al_2_O_3_, Na_2_O and K_2_O contents in experimental binder mixes.

Binder Mix	Al_2_O_3_	Na_2_O	K_2_O	Na_2_O + K_2_O
		[wt.%]		
BM1	4.17	0.75	5.04	5.79
BM2	3.09	0.87	5.03	5.90
BM3	4.14	0.76	5.05	5.81

**Table 12 materials-14-00041-t012:** Depth of needle penetration for the reference and experimental hydraulic road binder mixes.

Time [min]	Initial Setting Time Determined by Depth of Needle Penetration [mm]			
	Referential	BM1	BM2	BM3
80	-	0	0	1
90	-	1	1	2
100	3	1	1	2
105	-	1	1	3
110	-	2	1	4
115	-	3	2	5
120	6	4	2	6
125	-	5	2	7
130	-	6	2	7
140	-	7	3	9
150	-	9	6	10

**Table 13 materials-14-00041-t013:** Mean spillage values for the experimental hydraulic road binder mixes compared to the reference mix.

	Reference	BM1	BM2	BM3
Spill Diameter [mm]	152	148	146	164.5

**Table 14 materials-14-00041-t014:** Bulk density of the hardened binder specimens.

Specimens Sample	Bulk Density [kg/m^3^]		
	28 Days	56 Days	56 Days
Reference	2190 ± 18	2090 ± 12	2120 ± 7
BM1	2250 ± 23	2140 ± 16	2188 ± 8
BM2	2217 ± 3	2100 ± 8	2147 ± 11
BM3	2170 ± 5	2160 ± 10	2150 ± 8

**Table 15 materials-14-00041-t015:** Relative compressive strengths of hardened test specimens.

Specimen	Relative Compressive Strengths after Hardening		
	28 Days	56 Days	90 Days
BM1	1.19	1.14	1.15
BM2	1.09	1.17	1.03
BM3	0.98	1.23	1.08

## Data Availability

The data presented in this study are available on request from the corresponding author.

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
