# Peer review of "Incorporation of Cement Bypass Dust in Hydraulic Road Binder"

_materials, 2020, doi:10.3390/ma14010041_

Round 1
Reviewer 1 Report
The manuscript titled "Incorporation of Cement Bypass Dust in Hydraulic 2 Road Binder" describes an experimental study on the utilization of cement kiln by-pass dust as an added component in a hydraulic road binder.
The manuscript is very well written, the methods are described in detail and the relevant data regarding the materials are presented. The experiments were prepared and carried out thoroughly following standard procedures.
Just a minor observation, in line 240 unit seems to be missing.
In my opinion, the manuscript can be accepted for publication in Materials.
Reviewer 2 Report
The article is interesting and has a significant application potential. However, it has also a potential for improvement.
- The authors fail to define the scientific problem and demonstrate originality of the presented study. The research objective as described in lines 114-115, is strictly application- and implementation-oriented.
- Line 73 and below: Remove brand name of the producer.
- Section 2: remove brand names of producers and their products. Products shall be characterized by their constructional-material parameters only. Are the presented results producer-dependent? If yes, how it could be useful and interesting for the other readers?
- Figure 1 – remove
- Section 3: Statistical analysis of the presented results is missing.
- Section 4: Are the conclusions generalizing?
Reviewer 3 Report
The manuscript presents an experimental work aimed at using 10% CBD as a supplementary cementitious material for hydraulic road binder. There are following questions:
- In the literature review it is suggested to include a summary of other studies, compiling the applied methodology and the main conclusions. It shouldn’t just be a background description of CBD, and state‐of-art review performed should be strengthened.
- The authors should clarify the aim and the novelty of the study.
- The author should describe the basis for the determination of the relevant parameters for the design of mixture in this study.
- The results of the properties of fresh cement mixes should be further analyzed and mechanism discussion, and describe the influence of the physical and chemical properties of the material on this result.
- In Fig.3, 1, 2, 3 of the abscissa should be corrected to BM1, BM2 and BM3.
- The hardening properties of mixture should be compared with other studies.
Round 2
Reviewer 3 Report
I have no other comments.